

# Temperature affects the morphology and calcification of *Emiliania huxleyi* strains

Anaid Rosas-Navarro[1], Gerald Langer[2], and Patrizia Ziveri[1,3]

[1]Institute of Environmental Science and Technology (ICTA), Autonomous University of Barcelona (UAB), 08193 Bellaterra, Spain
[2]Department of Earth Sciences, Cambridge University, Downing St., Cambridge, CB2 3EQ, UK
[3]Catalan Institution for Research and Advanced Studies (ICREA), 08010 Barcelona, Spain

*Correspondence to:* A. Rosas-Navarro (anaid.rosas@uab.cat) and P. Ziveri (patrizia.ziveri@uab.cat)

**Abstract.** Three strains of *E. huxleyi* type A were grown under non-limiting nutrient and light conditions, at 10, 15, 20 and 25 °C. All three strains displayed similar growth rate versus temperature relationships, with an optimum at 20-25 °C. Elemental production (particulate inorganic carbon (PIC), particulate organic carbon (POC), total particulate nitrogen (TPN)), coccolith mass, coccolith size, and width of the tube elements cycle were positively correlated with temperature over the sub-optimum to

optimum temperature range. The correlation between PIC production and coccolith mass/size supports the notion that coccolith mass can be used as a proxy for PIC production in sediment samples. Increasing PIC production was positively correlated with the percentage of incomplete coccoliths in one strain only. Generally, coccoliths were heavier when PIC production was higher. This shows that incompleteness of coccoliths is not due to time shortage at high PIC production. Sub-optimal growth temperatures lead to an increase in the percentage of malformed coccoliths in a strain-specific fashion. Since in total only

six strains have been tested thus far, it is presently difficult to say whether sub-optimal temperature is an important factor causing malformations in the field. The most important parameter in biogeochemical terms, the PIC:POC, shows a minimum at optimum growth temperature in all investigated strains. This clarifies the ambiguous picture featuring in the literature.

## 1   Introduction

*Emiliania huxleyi* (Lohmann) Hay and Mohler, is a cosmopolitan (McIntyre and Bé, 1967; Brown, 1995), genetically diverse

(Medlin et al., 1996; Schroeder et al., 2005; Iglesias-Rodríguez et al., 2006; Hagino et al., 2011; Read et al., 2013), morphologically variable (Hagino et al., 2005; Hagino and Okada, 2006; Cubillos et al., 2007) marine photosynthetic and calcifying (Brownlee and Taylor, 2004) unicellular haptophyte algae species and the most abundant of the coccolithophores. It produces calcite ($CaCO_3$) plates called coccoliths which cover the cell. As a photosynthetic organism, *E. huxleyi* shifts the seawater carbonate system towards $[CO_2]$, but as a calcifier it shifts the seawater carbonate system towards $[CO_3^{2-}]$. Therefore, part of

the interest in *E. huxleyi* derives from its role in the global carbon cycle, traditionally in particular during the extensive blooms formed by this algae (Westbroek et al., 1993; Paasche, 2001), with potential consequences for the air–sea interchange of $CO_2$ (Robertson et al., 1994; Buitenhuis et al., 1996). Warming and freshening of the Bering Sea waters, associated with climate change, were found to trigger *E. huxleyi* blooms observed since the late 1970's in the area (Harada et al., 2012).



The particulate inorganic carbon (PIC) and the particulate organic carbon (POC) are used to calculate the PIC:POC ratio, which is an important variable both in terms of surface water carbonate chemistry and carbon export to depths (Findlay et al., 2011). The response of PIC and the POC production and their ratio in the prolific species *E. huxleyi* to temperature, is a necessary first step towards an understanding of its possible impact on global biogeochemical cycles.

The relationship of PIC production / PIC:POC and temperature in *E. huxleyi* is not clear. De Bodt et al. (2010) found that PIC production was higher at lower temperatures in a strain grown at 13 and 18 °C, while Sett et al. (2014) found the opposite in another strain grown at at 10, 15 and 20 °C. De Bodt et al. (2010) found higher PIC:POC ratios at lower temperatures for a strain of *E. huxleyi* and Gerecht et al. (2014) found a similar relationship for a strain of the species *Coccolithus pelagicus*. Sett et al. (2014), however, found a different relationship for the PIC:POC ratio in another strain of *E. huxleyi*, which is not supported

by the experiment of Langer et al. (2007) on the same strain. Feng et al. (2008) did not find differences in the PIC:POC ratio in another strain grown at 20 and 24 °C. These discrepancies between studies might stem from different experimental setups and a lacking knowledge of the optimum growth temperature or indeed strain-specific differences (Hoppe et al., 2011). Therefore it is necessary to test more than one strain for its temperature response under otherwise identical conditions. This we have done in the present study.

Apart from biogeochemical considerations, global warming might also be of interest in terms of the ecological success of coccolithophores. The latter was proposed to depend on coccolith morphology more than it does on PIC production (Langer et al., 2011). The effect of temperature on coccolith morphogenesis is evident in field observations (Bollmann, 1997; Ziveri et al., 2004) and is best assessed with respect to the optimum growth temperature in laboratory experiments. While the effect of supra-optimal temperature is unequivocally detrimental (Watabe and Wilbur, 1966; Langer et al., 2010), it is not clear whether

there is an effect of sub-optimal temperature at all (Watabe and Wilbur, 1966; Langer et al., 2010; De Bodt et al., 2010). A temperature increase in the sub-optimal range is probably what most coccolithophore clones will experience in the course of global warming (Buitenhuis et al., 2008; Langer et al., 2009; Heinle, 2014, this study), and therefore this temperature range is particularly interesting. In the present study we focus on coccolith morphology under sub-optimal temperature, doubling the amount of data currently available, and thereby clarifying whether sub-optimal temperatures can cause malformations.

## 2   Materials and methods

### 2.1   Pre-culture and batch culture experiments

Clonal cultures of *Emiliania huxleyi* were obtained from the Roscoff Culture Collection. We selected 3 strains of *E. huxleyi*, two from the Japanese coast in the North Pacific Ocean (RCC1710 –alternatively, NG1– and RCC1252 –alternatively, AC678 or MT0610E–) and a third strain of unknown origin named here IAN01. Additional information about the strains can be

found at: http://roscoff-culture-collection.org/. The culture media was sterile-filtered North Sea seawater (filtered through 0.2 µm pore size sterile filter cartridges) supplemented with nutrients (nitrates and phosphates), metals and vitamins, according to Guillard and Ryther (1962). Cell densities were determined using a Multisizer 3 Coulter Counter (Beckman-Coulter for particle characterization). To prevent significant changes in seawater carbonate chemistry maximum cell densities were limited to $\approx 1\times$



$10^5$ cells ml$^{-1}$ (e.g., Oviedo et al., 2014). We used a16/8 light/dark cycle, and an irradiance of $\approx 300\,\mu\text{mol} \cdot \text{photons} \cdot \text{s}^{-1}\text{m}^{-2}$. The three strains were grown for at least twenty generations.

The dilute batch culture experiments were conducted in triplicate, for the strains RCC1710 and RCC1252 at 10, 15, 20 and 25 °C of temperature, and for IAN01 at 15, 20 and 25 °C. The strains were grown in transparent sterilized 2.3 l glass bottles, in 2 liters of sea water. Cell density at inoculation was 500 cells ml$^{-1}$ to 1000 cells ml$^{-1}$, and at harvest was maximum $1 \times 10^5$ cells ml$^{-1}$. Harvesting was done nine hours after the onset of the light period.

Growth rate was calculated from exponential regression according to:

$$\mu = (\ln c_1 - \ln c_0)\Delta t^{-1}, \tag{1}$$

where $c_1$ and $c_0$ are the final cell concentration and the initial cell concentration, respectively, and $\Delta t$ is the duration of incubation in days. Averages of triplicates and SD were used in Tables and Figures (Table 1 and Fig. 1a).

## 2.2 Carbonate chemistry

Samples for total alkalinity (TA) measurements were sterile-filtered (0.2 μm pore size) and stored in 25 ml borosilicate flasks at 4 °C until measurements. TA was calculated from linear Gran plots (Gran, 1952) after potentiometric titration (in duplicate) (Bradshaw et al., 1981; Brewer et al., 1986).

Samples for dissolved inorganic carbon (DIC) were sterile-filtered (0.2 μm pore size) with gentle pressure using cellulose-acetate syringe filters and stored bubble-free at 4 °C in 5 ml borosilicate flasks until measurements. DIC was measured, in triplicate, using a Shimadzu TOC 5050A.

The carbonate system was calculated from temperature, salinity (32‰), TA, and DIC, using the the program CO2Sys (Lewis and Wallace, 1998), applying the equilibrium constants from Mehrbach et al. (1973), refitted by Dickson and Millero (1987). For an overview of carbonate chemistry conditions in all treatments, see Table 2.

## 2.3 Particulate organic and inorganic carbon, particulate nitrogen, calcite

Duplicate samples for the determination of total particulate carbon (TPC) and total particulate nitrogen (TPN), were filtered onto pre-combusted (500 °C; 12 h) 0.6 μm nominal pore-size glass fibre filters (Whatman GF/F), placed in pre-combusted petri dishes (500 °C; 12 h), oven dried (60 °C  24 h) and stored at -20 °C. Before analysis, TPC and TPN samples were stored in a desiccator. All samples were then measured on a Euro EA Analyser (Euro Vector).

Particulate inorganic carbon (PIC) was calculated measuring calcium content of samples with $3.6 \times 10^6$ *E. huxleyi* cells filtered onto 47 mm polycarbonate (PC) filters (0.8 μm pore size). PC filters were immersed overnight in an acid solution of 1% HNO$_3$ to dissolve calcite. Calcium was determined by analyzing an aliquot of the samples using an Inductively Coupled Plasma Mass Spectrometer (ICP-MS, Agilent model 7500ce). Cellular PIC was calculated from the molecular mass of calcite, using the following equations:

$$\text{PIC}_{\text{cell}^{-1}} = \frac{\text{PIC}_{\text{s}}}{c \cdot V_{\text{s}}}, \qquad \text{where} \quad \text{PIC}_{\text{s}} = \frac{[\text{Ca}^{2+}]_{\text{s}} \cdot 12.0107}{40.078}, \tag{2}$$



where $PIC_{cell^{-1}}$ is the cellular PIC (in pg), $PIC_s$ is the PIC sampled contained in the filter (in pg), $c$ is the cell concentration (in cells $l^{-1}$), $V_s$ is the volume sampled (in l), $[Ca^{2+}]_s$ is the calcium content in the sample (in pg), 12.0107 corresponds to the relative atomic mass of carbon, and 40.078 corresponds to the relative atomic mass of calcium. Particulate organic carbon (POC) was calculated as the difference between TPC and PIC. PIC, POC and TPN production ($P_{PIC}$, $P_{POC}$, $P_{TPN}$) were

estimated as the product of cellular PIC, POC or TPN, and growth rate. Calcite ($CaCO_3$) per cell (concomitant of PIC) can also be estimated, substituting in Eq. (2) the calcium carbonate molecular mass (100.0869) in place of the relative atomic mass of carbon. The ratio between PIC and POC (PIC:POC) and the ratio between POC and TPN (POC:TPN) were also calculated.

## 2.4 Coccolith morphology –by scanning electron microscopy

Thirty milliliters of culture were filtered onto polycarbonate filters (0.8 µm pore size) and dried at $60\,^{\circ}C$ for 24 hours. A

small portion ($\sim 0.7\,cm^2$) of each filter was mounted on an aluminium stub and coated with gold (EMITECH K550X Sputter Coater). Images were captured along random transects using a ZEISS-EVO MA10 scanning electron microscope (SEM).

*Emiliania huxleyi* SEM images were used to measure and categorize 300 coccoliths per sample (e.g., Langer et al., 2009); the coccoliths were on coccospheres. The tube width (width of the tube elements cycle) of each coccolith (Fig. 2c) was the average of the tube width measured on the two semi-minor axes (along the coccolith width) on the distal view of the coccolith.

Tube width measurements were manually taken using the program Gimp-2.8. Examples of the tube width variations in the three different strains are shown in Fig. 2. The 300 coccoliths were classified as normal, malformed or incomplete (e.g., Langer et al., 2011), as described in Table 3, with examples in Figs. 3 and 4.

## 2.5 Coccolith length and mass –by polarized light microscopy

10–30 ml of culture was filtered with $\sim 200$ mbar onto cellulose nitrate filters (0.2 µm pore size) and dried at $60\,^{\circ}C$ for 24

20    hours. A radial piece of filter was embedded and made transparent in immersion oil on microscope slides (e.g., Ziveri et al., 1995).

Images were taken at a magnification of 1000x with a Leica DM6000B cross-polarized light microscope (LM) equipped with a SPOT Insight Camera (e.g., Bach et al., 2012; Horigome et al., 2014). From 50 to 200 image frames from each sample were taken along radial transects and analyzed by the SYRACO software (Dollfus and Beaufort, 1999; Beaufort and Dollfus,

2004). A minimum of 300 coccolith images were automatically identified by the software and measured in pixels. The software also measures automatically the grey level for each pixel by a birefringence method based on the coccolith brightness when viewed in cross-polarized light (Beaufort, 2005). Coccolith length and mass were subsequently calculated from the pixels and from the measured grey level, respectively, following Horigome et al. (2014) and Beaufort (2005). Therefore, coccolith length was converted from pixels to micrometers, where 832 pixels correspond to 125 µm, and coccolith mass was converted from

grey level units to picograms, where 2275.14 grey level units were equivalent to 1 pg of calcite.





## 2.6 Statistics

For the three *E. huxleyi* strains together, an ANOVA (two-factor with replication) was used to test if a response variable (i.e. growth rate, element variables, morphological variables and mass) presented significant ($p < 0.05$) differences between the temperature treatments, to test if the effect was strain-independent or strain-specific ($p < 0.05$), and to test if there were significant differences in the interaction between treatment and strain ($p < 0.05$), so if the different strains respond similarly or not whether or not they were presenting differences between them. If the temperature effect was strain-specific, further ANOVA were used for pairs of strains.

If a response variable presented significant differences between the temperature treatments, and the variable also presented a significant strain-independent response to temperature, or at least the same response on two of the strains, the variable for the similar strains was analyzed with simple and multiple linear regressions, including $CO_2$ partial pressure ($p\text{CO}_2$), $CO_3^{2-}$ concentration and pH, in order to find the useful coefficients (*t*-statistics, $p < 0.05$) of the significant equation (*F*-test, $p < 0.05$) that would estimate the assessed variable value; e.g. the single or combined variables significantly estimating growth rate.

## 3 Results

### 3.1 Population growth

The three strains of *E. huxleyi* presented a stable growth rate (per day) that changed with temperature (Fig. 1a, Table 1), with significant differences between the temperature treatments ($F = 244.11$, $p = 0.000$). The strains RCC1710 and RCC1252 presented similar growth rates, not statistically different from one another ($F = 0.372$, $p = 0.550$). The IAN01 growth rate was significantly different from the other two *E. huxleyi* strains ($F = 4.53$, $p = 0.025$), but there was no significant difference in the interaction between treatment and strain ($F = 0.71$, $p = 0.597$), so the three strains behaved significantly similar. The optimum temperature for the three strains was 25 °C. When RCC1710 and RCC1252 were analyzed together, changes in growth rate only depended significantly on temperature (linear regression: $R^2 = 0.91$, $F = 229.58$, $p = 0.000$); the carbonate system variables (Table 2) did not increase much the coefficient of determination (maximum to an $R^2 = 0.92$) and none of them were significantly useful in predicting growth rate when used together with temperature (*t*-statistics: $p > 0.05$). According to Eq. (1), on the three strains, a minimum of one duplication per day was obtained from 15 to 27.5 °C.

### 3.2 Element measurements, ratios and production

Cellular PIC (and its concomitant calcite), POC and TPN ($\text{pg cell}^{-1}$) did not show a consistent trend related with temperature when comparing the three strains of *E. huxleyi* (Figs. 1b, e, h; Table 1). When cellular PIC and TPN response to temperature (from 15 to 25 °C) were statistically analyzed (ANOVA), significant differences were found between treatments ($F = 113.42$, $p = 0.000$ and $F = 36.52$, $p = 0.000$, respectively), but were not strain-independent ($F = 182.86$, $p = 0.000$ and $F = 33.32$, $p = 0.000$, respectively). Cellular POC, conversely, did not show significant differences between strains ($F = 1.71$, $p = 0.209$), but also did not show significant differences between the temperature treatments ($F = 0.09$, $p = 0.908$). Analyzing the three





strains independently, cellular PIC, POC, and TPN were significantly related with a carbonate system variable. However, there was no clear explanatory variable and responses were strain-specific.

In the three strains, production of PIC (and its concomitant calcite), POC and TPN ($\mathrm{pg\,cell^{-1}day^{-1}}$), showed a positive relationship with temperature (Figs. 1c, f, i; Table 1). Highest PIC and POC production was in general reached at $25\,°C$, except for RCC1710 that was reached at $20\,°C$. From the statistical analysis, PIC and POC production response to temperature, when comparing the three strains of *E. huxleyi* together, was significantly different between the temperature treatments ($F = 8.36$, $p = 0.003$) and the response was strain-independent ($F = 0.89$, $p = 0.428$). Highest TPN production was in general reached at $20\,°C$, except for RCC1252 that was reached at $25\,°C$. The latest was supported statistically, as TPN production response, with significant differences between temperature treatments ($F = 499.96$, $p = 0.000$), was strain specific ($F = 65.92$, $p = 0.000$) when comparing the three strains of *E. huxleyi* together, and yet still the strains RCC1710 and IAN01 presented a similar interaction between treatment and strain ($F = 3.52$, $p = 0.062$), thus the two strains had a similar behavior in the TPN production response despite the different values between the strains ($F = 19.02$, $p = 0.000$).

Changes in PIC production on the three strains of *E. huxleyi* mostly depended on temperature (linear regression: $R^2 = 0.89$, $F = 217.36$, $p = 0.000$); $p\mathrm{CO_2}$ with $[\mathrm{CO_3^{2-}}]$, when used together with temperature, just increased slightly the coefficient of determination ($R^2 = 0.93$). Changes in POC production on the three strains of *E. huxleyi* only depended significantly on temperature (linear regression: $R^2 = 0.85$, $F = 157.71$, $p = 0.000$).

The PIC:POC ratio decreased from 10 to $20\,°C$ in the three strains of *E. huxleyi* (Fig. 1d). POC was higher than PIC only in the strains RCC1710 and IAN01 at $20\,°C$. From the statistical analyses, the only significant similitude obtained was in the interaction between treatment and strain for RCC1252 and IAN01 ($F = 2.12$, $p = 0.163$), which means that the PIC:POC ratio behaves similarly towards temperature in these two strains.

The POC:TPN ratio (Fig. 1h) relationship with temperature was strain-specific ($F = 9.59$, $p = 0.001$). The differences between the temperature treatments were significant ($F = 16.95$, $p = 0.000$). There were no significant differences between the strains RCC1710 and RCC1252 ($F = 2.71$, $p = 0.119$), in which lowest POC:TPN ratio was found at $10\,°C$, however there were significant differences in the interaction between treatment and strain ($F = 3.52$, $p = 0.039$), as observed in the different temperatures at which maximum POC:TPN ratio were found for each strain (20 and $25\,°C$, respectively). The strain IAN01 showed a much different relationship with temperature, with a minimum POC:TPN ratio found at $20\,°C$.

### 3.3 Coccolith morphology and mass

Although there was great variation between replicates, mean tube width of coccoliths (Fig. 5a, Table 4) presented a positive trend with temperature, independently of the strain of *E. huxleyi* ($F = 1.73$, $p = 0.204$). Changes in tube width on the three strains of *E. huxleyi* only depended on temperature (linear regression: $R^2 = 0.47$, $F = 28.09$, $p = 0.000$); $p\mathrm{CO_2}$ and $[\mathrm{CO_3^{2-}}]$ did not increase much the coefficient of determination ($R^2 = 0.51$) and none of them were significantly useful in predicting tube width when used together with temperature (*t*-statistics: $p > 0.05$).

Coccolith length (Fig. 5b, Table 4) showed a positive trend with temperature, specially on strains RCC1252 and IAN01. The positive trend in strain RCC1710 was not so clear, however, minimum length was also found at $10\,°C$ and maximum length also





at 25 °C. Strains RCC1252 and IAN01 were analyzed together in a multiple linear regression analysis, as they did not present significant differences between them ($F = 2.12$, $p = 0.171$); temperature gave the highest coefficient of determination ($R^2 = 0.62$, $F = 24.03$, $p = 0.000$) and was the only useful coefficient in estimating coccolith length, when making any combination with $pCO_2$, $[CO_3^{2-}]$ or pH. The strain RCC1710 was analyzed independently of the other two strains: temperature presented a low and not significant coefficient of determination ($R^2 = 0.28$, $F = 3.55$, $p = 0.092$), instead, pH presented the highest coefficient of determination ($R^2 = 0.65$, $F = 16.87$, $p = 0.002$).

Regardless of the strain, coccolith calcite mass (Fig. 5c, Table 4) showed a positive trend with temperature; significant differences were found between treatments ($F = 35.59$, $p = 0.000$) and no significant differences were found in the interaction between treatment and strain ($F = 2.53$, $p = 0.08$). The strains RCC1252 and IAN01 were analyzed together as they did not show significant differences between them ($F = 0.65$, $p = 0.425$). Temperature presented the highest coefficient of determination for RCC1252 and IAN01 ($R^2 = 0.75$, $F = 45.93$, $p = 0.000$) and also for RCC1710 ($R^2 = 0.87$, $F = 58.58$, $p = 0.000$), and adding other coefficients was not significantly useful in estimating coccolith mass. On average, coccolith mass increased with temperature $\sim 2.2$ times from 10 to 25 °C, $\sim 1.5$ times from 15 to 25 °C, and $\sim 1.2$ times from 20 to 25 °C; on average, coccolith mass increased 1.28 times (or 0.45 pg) each 5 °C.

The percentage of malformed coccoliths per sample (Fig. 6a, Table 4), did not show a consistent trend with temperature when comparing the three strains of *E. huxleyi* ($F = 113.21$, $p = 0.000$). Only one strain (RCC1252) presented a significantly higher percentage at the lowest experimented temperature.

Higher percentages of incomplete coccoliths were found at 20 or 25 °C (Fig. 6b, Table 4). ANOVA results showed that, between the three strains, there were no significant differences only between the strains RCC1252 and IAN01 ($F = 0.06$, $p = 0.810$) and their interaction between treatment and strain ($F = 2.33$, $p = 0.139$), though in this case (analyzed from 15 to 25 °C) there were also no significant differences between the temperature treatments ($F = 3.78$, $p = 0.053$).

## 4 Discussion

### 4.1 Elemental production and incomplete coccoliths

All three *E. huxleyi* strains investigated here displayed similar growth rate versus temperature relationships, with an optimum at 20-25 °C (Fig. 1a). This is a typical range for many *E. huxleyi* strains (Watabe and Wilbur, 1966; Van Rijssel and Gieskes, 2002; Sorrosa et al., 2005; De Bodt et al., 2010; Langer et al., 2009). Also not untypical, elemental production (PIC, POC, TPN) increased with temperature over the sub-optimum to optimum temperature range (Fig. 1, Langer et al. (2007); Sett et al. (2014)). It is intuitive that, approaching optimum, higher temperature increases elemental production, because biochemical rates are temperature dependent. It is also intuitive that the percentage of incomplete coccoliths should increase with higher $P_{\mathrm{PIC}}$, as indeed observed in RCC1710 (Fig. 6b). The idea underlying this intuition is that less time is taken to produce one coccolith and that the production process is stopped before the coccolith is fully formed. A comparison of RCC1710 and RCC1252 shows how wrong this idea is (Table 6). The percentage of incomplete coccoliths increases in the former only. While it is true that coccolith production time in RCC1710 decreases from 29 min at 10 °C to 22 min at 25 °C, this decrease is even





more pronounced in RCC1252 (from 85 min to 23 min). Hence RCC1252 should show a steeper increase in incompleteness than RCC1710. This is not the case. Please note that the increase in incompleteness in RCC1252 (Fig. 6b), is not significant, because the increase is well below $10\%$ and the error bars overlap (see also Langer et al. (2013) for a discussion of this criterion). Another piece of evidence which does not fit the "premature release of coccoliths because of time shortage" idea is

5 that both RCC1710 and RCC1252 manage to produce heavier coccoliths in a shorter time at higher temperature (Table 4 and Table 6). We do not know why the stop-signal for coccolith growth is affected by temperature in RCC1710. Nothing is known about the biochemical underpinning of that stop-signal, so it is unfortunately impossible to speculate about the mechanism of a temperature effect. It was, however, argued that the processes involved in the stop signal are different from those producing teratological malformations (Young and Westbroek, 1991; Langer et al., 2010, 2011). This is supported by our data, because

there is no correlation between incompleteness and malformations (Fig. 6). We will discuss malformations in section 4.3.

Interestingly coccolith mass is positively correlated with temperature (and $P_{\mathrm{PIC}}$) in all strains tested here. The positive correlation of coccolith mass and $P_{\mathrm{PIC}}$ was also observed by Bach et al. (2012) in a carbonate chemistry manipulation experiment and is the basis of using coccolith mass as a proxy for $P_{\mathrm{PIC}}$ (Beaufort et al., 2011). This is an interesting option, because in field samples coccolith mass might be a promising indicator of $P_{\mathrm{PIC}}$. There are only few proxies available to reconstruct

past coccolithophore $P_{\mathrm{PIC}}$, the traditional one being the calcite Sr/Ca, established at the turn of the millennium (Stoll and Schrag, 2000). Analysing Sr/Ca, however, requires either a sizable sample or comparatively sophisticated Secondary Ion Mass Spectrometry (SIMS) measurements (Stoll et al., 2007; Prentice et al., 2014). Recently, coccosphere diameter and coccolith quota were introduced as growth rate proxies (Gibbs et al., 2013). However, complete coccospheres are the exception rather than the rule in sediment samples, so it is important to have a proxy based on individual coccoliths. Hence coccolith mass and

size (which are correlated, Fig. 5, Table 4), are an option which it is worthwhile exploring in the future.

### 4.2 *Emiliania huxleyi* PIC:POC response

As detailed in the introduction there is considerable variability in the PIC:POC response of *E. huxleyi* to temperature changes. This variability cannot be traced back to strain-specific features, but might partly reflect the fact that different temperature ranges were investigated, mostly without the knowledge of the optimum temperature. Also other experimental conditions, such

as light intensity and nutrient concentrations, varied and might have played a role (Hoppe et al., 2011). In this study we ran three strains under identical conditions, and, for the first time, are presented with a coherent picture. All three strains display a bell shaped curve with lowest PIC:POC close to the optimum growth temperature (Fig. 1d). Although our data on the right-hand side of the PIC:POC minimum are not conclusive for RCC1252, the bell shaped curve is discernible in the latter strain. This finding seems to fit data on another *E. huxleyi* strain (De Bodt et al., 2010) and on *C. pelagicus* (Gerecht et al., 2014).

This comparison is, however, not straightforward since both studies employed two temperatures only without determining the optimum temperature. Be that as it may, based on our data, we might conclude that *E. huxleyi* tends to show the lowest PIC:POC close to its optimum growth temperature. In the context of global warming, that would mean that in the future, *E. huxleyi* and maybe coccolithophore PIC:POC will tend to decrease because most strains live at sub-optimal temperatures in the field. This trend might be pronounced because global warming is accompanied by lower surface water nutrient levels and ocean




acidification (Cermeno et al., 2008; Doney et al., 2009). All these changes apparently cause a decrease in *E. huxleyi*'s PIC:POC (our data, Hoppe et al. (2011), Oviedo et al. (2014)). A marked decline in coccolithophore PIC:POC will have implications for long term carbon burial and might even affect surface water carbonate chemistry on short timescales (Barker et al., 2003; Ridgwell and Zeebe, 2005; Cermeno et al., 2008).

## 4.3 Coccolith malformations

Coccolith malformations, i.e. disturbances of the coccolith shaping machinery, occur in both field and culture samples, but usually more so in the latter (Langer et al., 2006, 2013). The causes of malformations are only partly known. In cultured samples, artificial conditions (not present in the field) play a role, inducing the surplus of malformations compared to field samples (Langer et al., 2013; Ziveri et al., 2014). However, in the field malformations do occur, and sometimes in considerable percentages (Giraudeau et al., 1993; Ziveri et al., 2014). The environmental conditions leading to elevated levels of malformations have long since been disputed. Besides nutrient limitation (Honjo, 1976), temperature and carbonate chemistry are conspicuous candidates. In a seminal experimental study it was shown that moving away from the optimal growth temperature increases malformations in *E. huxleyi* (Watabe and Wilbur, 1966). This result was confirmed for higher than optimum temperature in another strain (Langer et al., 2010), but could not be confirmed for sub-optimal temperature in two strains (De Bodt et al., 2010; Langer et al., 2010). The sub-optimal temperature range is of particular interest because most clones live at sub-optimal temperatures in the field. Here we investigated sub-optimum to optimum temperatures in three further strains. While RCC1710 showed no change in the percentage of malformations and IAN01 featured a shallow gradual increase from 25 to 15 °C, RCC1252 was insensitive over the latter range, but displayed a steep increase in malformations at 10 °C (Fig. 6). Based on our own and the literature data, we conclude that the sub-optimal temperature effect on morphogenesis is strain-specific. Can we see a pattern in this strain specificity? It is intriguing that *E. huxleyi* clones fall into two distinct groups characterized by their temperature preference, the warm-water and the cool-water group (Hagino et al., 2011). Of the strains analysed for morphology the following belong to the warm-water group: BT-6 (Watabe and Wilbur, 1966), RCC1252, and maybe RCC1238 (Langer et al., 2010). The latter was unfortunately not included in the study by Hagino et al. (2011). Since these strains display different responses to temperature, their being part of the warm-water group does unfortunately not help finding common features of sensitive strains. However, only few strains were studied so far and it might be worthwhile testing a statistical number from the warm-water and the cool-water group.

## 5 Conclusions

1) Temperature, PIC production, coccolith mass, and coccolith size are positively correlated. Since the positive correlation between coccolith mass and PIC production was observed in response to seawater carbonate chemistry changes as well (Bach et al., 2012), it can be hypothesized that coccolith mass might be a good proxy for PIC production independent of the environmental parameter causing the change in PIC production.





2) Sub-optimal growth temperature was identified as one of the potential causes of coccolith malformations in the field. Since the effect of sub-optimal temperature on coccolith morphogenesis is strain-specific, a statistically relevant number of strains has to be tested in order to clarify whether this effect is indeed ecologically relevant.

3) We consistently showed for the first time that *E. huxleyi* features a PIC:POC minimum under optimum growth temperature. Taken together with literature data this finding suggests that global environmental change will lead to a marked decrease in PIC:POC of *E. huxleyi* and possibly coccolithophores as a group.

*Acknowledgements.* This work was funded by the European Union's Seventh Framework Programme under grant agreement 265103 (project MedSeA), the European Research Council (ERC grant 2010-NEWLOG ADG-267931 HE) and the Generalitat de Catalunya (MERS, 2014 SGR - 1356). AR also acknowledge the "MECD/SGU/DGPU, Programa Estatal de Promoción del Talento y su Empleabilidad" (Becas FPU) of the MINECO, Spain. AR thanks the technical and personal support from researchers and technicians at the Alfred Wegener Institute for Polar and Marine Research (AWI) where the laboratory experiments were done. AR thanks Michael Grelaud for his advise on the use of the software SYRACO.



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





**Figure 1.** Results at different temperatures. Growth rate (a) (extra temperatures from pre-experiments are included and shown as empty symbols); cellular PIC and its concomitant calcite (b), POC (e) and TPN (h) content; PIC (c), POC (e) and TPN (i) production (linear trendlines and r-squared values are shown); PIC:POC ratio (d) and POC:TPN ratio (g). Standard deviations of the triplicate experiment results are shown. Three different strains of *E. huxleyi* were used.




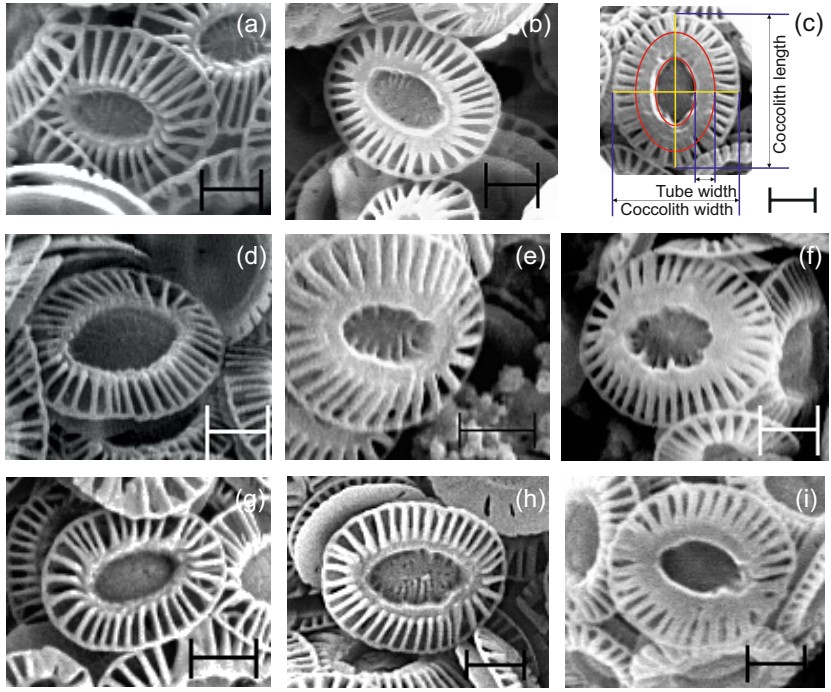

**Figure 2.** Examples of tube width variations observed in *Emiliania huxleyi* RCC1710 (a-c), RCC1252 (d-f), and IAN01 (g-i) coccoliths. Tube width (c) was measured along the two semi-minor axes (along the coccolith width) of each coccolith and averaged. Scale bar equal to 1 $\mu$m.

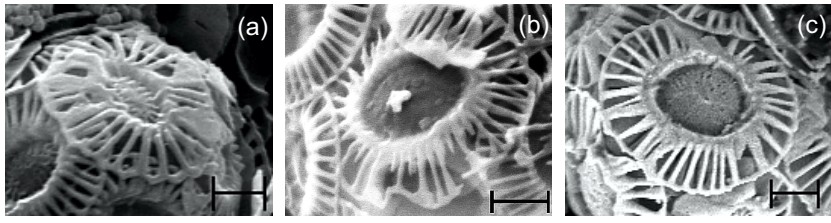

**Figure 3.** Examples of malformed coccoliths found in *Emiliania huxleyi* RCC1710 (a), RCC1252 (b), and IAN01 (c). Scale bar equal to 1 $\mu$m.




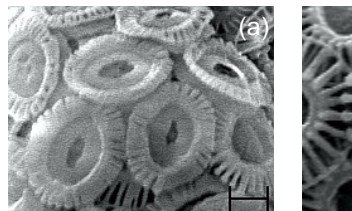
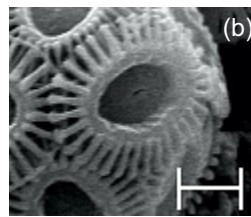
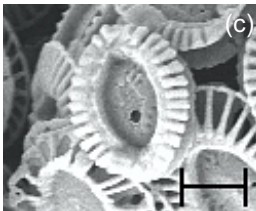

**Figure 4.** Examples of incomplete coccoliths of *Emiliania huxleyi* RCC1710 (a), RCC1252 (b), and IAN01 (c). Scale bar equal to 1 $\mu$m.

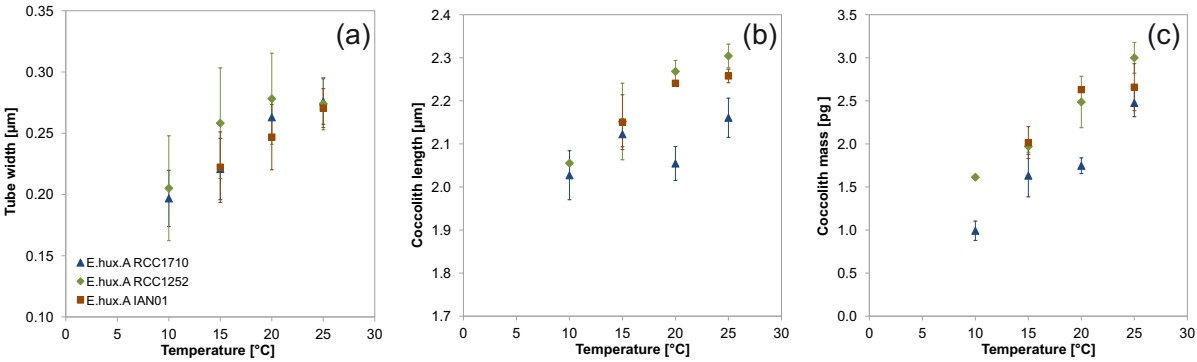

**Figure 5.** Changes in coccolith morphometry (a and b) and mass (c), at different temperatures. Standard deviations of the triplicate experiment results are shown. Three different strains of *E. huxleyi* were used.

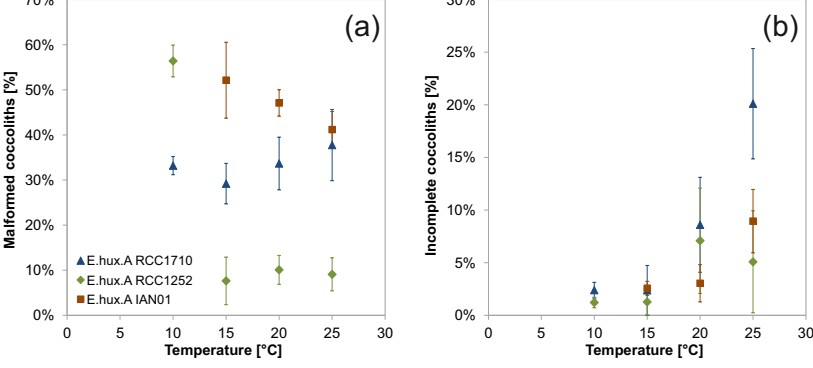

**Figure 6.** Percentage of malformed (a) and incomplete (b) coccoliths, in three *E. huxleyi* strains grown at different temperatures. Standard deviations of the triplicate experiment results are shown.



**Table 1.** Growth rate and cellular PIC, POC, and TPN content and production, of the three strains of *E. huxleyi* at different temperatures. Standard deviation of the triplicates in parentheses.

| Strain | $T$ [°C] | Growth rate ($\mu$) | PIC [pg cell$^{-1}$] | POC [pg cell$^{-1}$] | TPN [pg cell$^{-1}$] | $P_{PIC}$ [pg cell$^{-1}$ day$^{-1}$] | $P_{POC}$ [pg cell$^{-1}$ day$^{-1}$] | $P_{TPN}$ [pg cell$^{-1}$ day$^{-1}$] |
|---|---|---|---|---|---|---|---|---|
| RCC1710 | 6.5 | 0.19 | | | | | | |
| RCC1710 | 10 | 0.26 (0.00) | 15.31 (0.15) | 8.91 (0.29) | 1.54 (0.07) | 3.98 (0.03) | 2.32 (0.08) | 0.40 (0.01) |
| RCC1710 | 15 | 0.75 (0.01) | 14.07 (0.40) | 9.90 (0.11) | 1.47 (0.01) | 10.55 (0.41) | 7.42 (0.16) | 1.10 (0.01) |
| RCC1710 | 20 | 1.15 (0.02) | 11.47 (0.09) | 12.05 (0.79) | 1.71 (0.06) | 13.16 (0.15) | 13.82 (0.63) | 1.98 (0.04) |
| RCC1710 | 25 | 1.24 (0.01) | 10.80 (0.24) | 9.30 (0.80) | 1.38 (0.04) | 13.34 (0.33) | 11.48 (0.99) | 1.70 (0.06) |
| RCC1710 | 27.5 | 1.04 | | | | | | |
| RCC1710 | 30 | 0.23 | | | | | | |
| RCC1252 | 6.5 | 0.18 | | | | | | |
| RCC1252 | 10 | 0.26 (0.04) | 8.29 (0.49) | 6.35 (0.11) | 1.16 (0.03) | 2.15 (0.39) | 1.64 (0.23) | 0.30 (0.04) |
| RCC1252 | 15 | 0.73 (0.00) | 9.92 (0.32) | 8.64 (0.29) | 1.34 (0.03) | 7.22 (0.23) | 6.29 (0.22) | 0.97 (0.02) |
| RCC1252 | 20 | 1.15 (0.14) | 9.89 (0.28) | 8.75 (0.71) | 1.35 (0.07) | 12.01 (0.74) | 9.99 (1.13) | 1.56 (0.26) |
| RCC1252 | 25 | 1.22 (0.02) | 12.20 (0.21) | 10.19 (0.75) | 1.41 (0.02) | 14.84 (0.38) | 12.39 (0.86) | 1.72 (0.02) |
| RCC1252 | 27.5 | 1.02 | | | | | | |
| RCC1252 | 30 | 0.00 | | | | | | |
| IAN01 | 6.5 | 0.12 | | | | | | |
| IAN01 | 15 | 0.81 (0.01) | 10.18 (0.30) | 9.89 (0.43) | 1.47 (0.08) | 8.20 (0.19) | 7.97 (0.30) | 1.18 (0.06) |
| IAN01 | 20 | 1.17 (0.00) | 8.12 (0.21) | 8.95 (0.43) | 1.75 (0.09) | 9.46 (0.25) | 10.43 (0.51) | 2.04 (0.11) |
| IAN01 | 25 | 1.32 (0.03) | 11.21 (0.36) | 9.95 (0.11) | 1.46 (0.01) | 14.84 (0.49) | 13.17 (0.22) | 1.94 (0.03) |
| IAN01 | 27.5 | 1.01 | | | | | | |
| IAN01 | 30 | -0.11 | | | | | | |




**Table 2.** The carbonate system. Standard deviation of the triplicates in parentheses.

| Strain | $T$ [°C] | TA [µmol kg$^{-1}$] | DIC [µmol kg$^{-1}$] | pH (totalscale) | $p$CO$_2$ [µatm] | HCO$_3^-$ [µmol kg$^{-1}$] | CO$_3^{2-}$ [µmol kg$^{-1}$] | omega calcite |
|---|---|---|---|---|---|---|---|---|
| RCC1710 | 10 | 2138 (23) | 2012 (3) | 7.95 (0.07) | 482 (74) | 1893 (14) | 98 (15) | 2.38 (0.36) |
| RCC1710 | 15 | 2167 (14) | 2023 (12) | 7.92 (0.01) | 530 (13) | 1893 (11) | 111 (3) | 2.69 (0.07) |
| RCC1710 | 20 | 2291 (25) | 2110 (4) | 7.92 (0.06) | 571 (84) | 1953 (19) | 139 (18) | 3.39 (0.45) |
| RCC1710 | 25 | 2306 (24) | 2123 (7) | 7.86 (0.03) | 688 (55) | 1961 (4) | 142 (11) | 3.51 (0.28) |
| RCC1252 | 10 | 2249 (8) | 2095 (12) | 8.02 (0.03) | 427 (30) | 1959 (16) | 117 (6) | 2.84 (0.15) |
| RCC1252 | 15 | 2219 (57) | 2065 (6) | 7.94 (0.12) | 533 (136) | 1925 (21) | 119 (32) | 2.90 (0.78) |
| RCC1252 | 20 | 2212 (20) | 2043 (15) | 7.91 (0.01) | 571 (10) | 1896 (11) | 129 (4) | 3.15 (0.09) |
| RCC1252 | 25 | 2229 (8) | 2052 (10) | 7.85 (0.04) | 670 (64) | 1896 (19) | 137 (11) | 3.37 (0.26) |
| IAN01 | 15 | 2206 (9) | 2064 (16) | 7.92 (0.02) | 551 (33) | 1932 (19) | 111 (4) | 2.70 (0.11) |
| IAN01 | 20 | 2249 (28) | 2106 (6) | 7.84 (0.05) | 698 (86) | 1969 (5) | 115 (14) | 2.80 (0.34) |
| IAN01 | 25 | 2243 (2) | 2066 (4) | 7.85 (0.01) | 677 (13) | 1910 (5) | 137 (2) | 3.37 (0.05) |

**Table 3.** Morphological categorization of coccoliths (from SEM images) of *Emiliania huxleyi* used in this study.

| Morphological category | Description |
|---|---|
| Normal | Regular coccolith in shape, with well-formed distal shield elements aligned forming a symmetric rim. Considered normal when nil or only two malformations were present. |
| Malformed | Irregular coccolith in shape or size of individual elements and a general reduction in the degree of radial symmetry shown; teratological malformation (Young and Westbroek, 1991). Considered malformed when three or more malformations were present in the coccolith. |
| Incomplete | Coccolith with variations in its degree of completion according to its normal growing order, with no malformations. Primary calcification variation (Young, 1994). |





**Table 4.** Coccoliths morphology and mass. Standard deviation of the triplicates is shown in parentheses.

| Strain | $T$ | Tube width | Coccolith length | Coccolith mass | Malformed | Incomplete |
|--------|-----|-----------|------------------|----------------|-----------|------------|
|        | [°C] | [μm] | [μm] | [pg] | [%] | [%] |
| RCC1710 | 10 | 0.20 (0.02) | 2.03 (0.06) | 0.99 (0.11) | 33.18 (2.02) | 2.39 (0.75) |
| RCC1710 | 15 | 0.22 (0.03) | 2.12 (0.03) | 1.63 (0.25) | 29.19 (4.50) | 2.38 (2.36) |
| RCC1710 | 20 | 0.26 (0.02) | 2.05 (0.04) | 1.75 (0.09) | 33.66 (5.85) | 8.60 (4.51) |
| RCC1710 | 25 | 0.28 (0.02) | 2.16 (0.05) | 2.48 (0.16) | 37.75 (7.90) | 20.10 (5.24) |
| RCC1252 | 10 | 0.21 (0.04) | 2.06 (0.00) | 1.61 (0.00) | 56.39 (3.54) | 1.22 (0.51) |
| RCC1252 | 15 | 0.26 (0.05) | 2.15 (0.09) | 1.97 (0.07) | 7.65 (5.29) | 1.28 (1.25) |
| RCC1252 | 20 | 0.28 (0.04) | 2.27 (0.03) | 2.49 (0.30) | 10.09 (3.21) | 7.09 (5.01) |
| RCC1252 | 25 | 0.27 (0.02) | 2.30 (0.03) | 3.00 (0.18) | 9.09 (3.67) | 5.08 (4.85) |
| IAN01 | 15 | 0.22 (0.03) | 2.15 (0.06) | 2.02 (0.19) | 52.13 (8.41) | 2.58 (0.66) |
| IAN01 | 20 | 0.25 (0.03) | 2.24 (0.00) | 2.63 (0.00) | 47.09 (2.92) | 3.05 (1.78) |
| IAN01 | 25 | 0.27 (0.02) | 2.26 (0.02) | 2.66 (0.27) | 41.18 (4.01) | 8.95 (3.01) |

**Table 5.** Strain-independent and strain-specific responses of *E. huxleyi* to temperature, found in the three strains of this study.

| Strain-independent responses | Strain-specific responses |
|---|---|
| – Growth rate optimum temperature was 25 °C. | – Cellular PIC, POC and TPN (pg per cell). |
| – Highest PIC, POC, and TPN production values were found at 20 or 25 °C. | – POC:TPN ratio. However, in the two strains tested at 10 °C (RCC1710 and RCC1252), the POC:TPN ratio was lowest at 10 °C. |
| – The PIC:POC ratio decreased from 10 to 20 °C. | |
| – Tube width increased with temperature, from ∼ 0.20 μm at 10 °C to ∼ 0.27 μm at 25 °C. | – Percentage of malformed coccoliths per sample. |
| – Maximum coccolith length was found at 25 °C. | – Percentages of incomplete coccoliths. |
| – Coccolith mass increased with temperature (∼ 2.2 times from 10 to 25 °C, ∼ 1.5 times from 15 to 25 °C, and ∼ 1.2 times from 20 to 25 °C; on average, 0.45 pg each 5 °C). | – Coccolith length, although in strains RCC1252 and IAN01 was positively correlated with temperature. |



**Table 6.** Coccolith production time. Lith: coccolith, *d*: day, *h*: hour, min: minutes.

| Strain | $T$ [°C] | pgPIC $\cdot$ lith$^{-1}$ | Lith $\cdot$ cell$^{-1}$ | Lith $\cdot$ cell$^{-1} \cdot d^{-1}$ | Lith $\cdot$ cell$^{-1} \cdot h^{-1}$ | Min $\cdot$ lith$^{-1}$ | pgPIC $\cdot h^{-1}$ |
|---|---|---|---|---|---|---|---|
| RCC1710 | 10 | 0.12 | 128 | 33 | 2.1 | 29 | 0.25 |
| RCC1710 | 25 | 0.30 | 36 | 45 | 2.8 | 22 | 0.84 |
| RCC1252 | 10 | 0.19 | 43 | 11 | 0.7 | 85 | 0.13 |
| RCC1252 | 25 | 0.36 | 34 | 41 | 2.6 | 23 | 0.93 |