# Peer review of "Temperature affects the morphology and calcification of *Emiliania* huxleyi strains"

_Biogeosciences, 2015_

## Referee Comment (RC1) · Anonymous Referee #1 · 27 Jan 2016

General comments:

The authors' address the effects of temperature on Emiliania huxleyi's morphology and calcification. Due to increased anthropogenic emissions of greenhouse gases, combined land and sea surface temperature has risen and is expected to continue to increase. Its effects on calcifying phytoplankton are confounded since there isn't enough data to conciliate interpretations and establish the potential of malformations as proxy. The analysis of more strains manipulated similarly is an interesting approach. The observed correlations between coccolith mass and PIC production as well as data on the percentage of incomplete coccoliths in relation to temperature provides information that can potentially be used to understand the past.

The paper would benefit from better structured abstract and additional information in

the discussion, namely the authors might want to briefly discuss coccolith function and explain with more detail their formation process, since malformations might occur at different steps of the latter.

Although the temperature range used is physiologically interesting it is broader than expected for the year 2100. Therefore, this should be clear in the discussion.

Finally, the manuscript seems to follow reliable experimental procedures and design and is overall well written.

Specific comments and technical corrections:

Abstract

The abstract would benefit from a statement with the importance and aim of the study as well as a clear conclusion.

Page 1, line 20 - Replace "E. huxleyi" with " Emiliania huxleyi ".

Page 1, line 28 – Increasing PIC production (Figure 1C) was only positively correlated with the percentage of incomplete coccoliths in one strain? Do the authors mean significantly?

Page 2, line 1-3 – It could be related to time shortage of other steps of the coccolith formation.

Page 2, line 8 – Clarify the final sentence.

Introduction Page 3, line 9-10 – Correct and clarify.

Page 3, line 10-14 – Long sentence. Improve ". . . traditionally in particular. . .".

Page 3, line 14-16 – The sentence should include information concerning stratification, since it is a relevant point in the argument. Moreover, it needs to be clearer.

Page 3, line 17-18 – Improve the sentence.

Page 3, line 19 – Replace "...PIC and the POC production..." with "...PIC and POC production..." and improve sentence.

Page 4, line 12-13– Clarify the sentence.

Material and methods

When describing units, like "cells.ml-1", I would remove the ".".

Page 5, line 8 – Explain "alternatively".

Page 5, line 10 – Replace "North Sea seawater" with "North Sea water".

Page 5, line 11 – Is it relevant to state "filter cartridges"? Perhaps the filter composition could be more interesting.

Page 5, line 11-12 – Replace "nitrates and phosphates" with "nitrate and phosphate".

Page 6, line 8 and 13– For how long were the samples stored before being measured?

Page 6, line 17– Remove "," after TA

Page 6, line 22– I would add a "and" before "calcite".

Page 7, line 2– Storing the samples in a desiccator is important to dry the samples before analysis, correct? How long were they stored in the desiccator?

Page 8, line 18– Start the sentence with text.

Results

Results could be better organized and more fluid. Reference to effects of the carbonate system could be pooled in one / two paragraphs.

Page 10, line 12– For which temperature(s)?

Page 10, line 25– Correct "pgcell-1", "-1" should be superscript.

Page 11, line 7-9– It is not clear to which carbonate system variable it is being referred

here. It would be useful to have a short explanation on the material and methods section concerning the reasoning for following the carbonate system in the experiment. Is it part of your hypothesis? Moreover, the legend of Table 2 should state whether the values are initial, final or averages of the incubations.

Page 14, line 10-15- The paragraph should be clearer.

Discussion

Page 14, line 19– In spite of being referred in the text, this title does not refer growth rate.

Page 14, line 21– I would replace "versus" with "in relation to".

Page 14, line 22-23– Specify how many strains and add their isolation sites or areas.

Page 15, line 6- Why do you only show data for 2 strains and 2 temperatures?

Page 15, line 5- Add reference to support the statement.

Page 15, line 6- Percentage of incomplete coccoliths is higher under 20-25°C than 10-15°C in RCC1252. Thus, if one would consider the temperatures used to determine the coccolith production time, both strains would show similar trends, or not? The lack of significance precludes a strong conclusion concerning this parameter.

Page 15, line 10- What about coccolith mass?

Page 16, line 14- The authors refer a previous study (Sett et al., 2014) that showed a different relation between PIC:POC and temperature in the introduction section. This should be referred in the discussion section.

Page 17, line 6- Add references supporting ". . . most strains live at sub-optimal temperatures in the field.".

Page 17, line 10-13- Clarify what is meant with "short timescales".

Page 17, line 19-20- Specify which artificial conditions could play a role in producing

malformations of coccoliths.

Page 18, line 13-15- One of the strains tested in this study is considered to belong to the warm-water group while the others not. Why is it referred in the text and how can it affect the observed responses to increasing temperatures?

References

Young and Westbroek, 1991 cited on page 15 line 21 is missing in the references. Several references have typos, namely missing italics, incorrect formatting of the 2 of CO2 (should be subscript) and accents.

Page 30, line 7– Incorrect date, it should be 1993.

Legends

Table 5 and Figure 2, 3, 4- Species should be in italic and presented in a consistent form, either E. huxleyi or Emiliania huxleyi.

Figure 1, page 37, line 4- missing a space between "(E) and".

Tables

Table 5- The percentage of incomplete coccoliths is considered a strain-specific response, why? The author's should clarify the choice of responses in the legend of the table. Was it based on significance?

Figures

Figure 1, 5 and 6 are hard to analyze due to size / resolution. Labels (A, B. . .) of the figures and text should follow the same formatting. Figures that do not start with 0, should show an interruption on the axis. Finally, when the unit is in the axis title it is not necessary to have it close to the numbers (see for instance Figure 6).

---

## Referee Comment (RC2) · Anonymous Referee #2 · 15 Feb 2016

Emiliania huxleyi is an important dominant species of coccolithophore. The species has been well known by its great blooming. The event has large impact on the carbon cycle of marine environment because the organism leads both photosynthesis and calcification simultaneously during this blooming event. This study attempts to evaluate the change of the production of coccolith and organic matters under variable temperatures and multiple properties through the laboratory culture experiment. The efforts will bring the useful knowledge to biogeochemists, geochemists, micropalaeontologists, oceanographers and other audience from earth science. I would like to strongly support the authors' challenge of evaluation by experimental set up.

Questions: I would like to recommend authors to show the reason why the three strains are chosen. This species widely distributed from tropical to sub-polar region. If the strains originated from cooler environment were chosen, the opposite relationships

may be shown between production and temperature. The environmental adaptation might be important to include consideration of the result. The other ecological information of them are also valuable if they are available.

P2L30ãĂĂThe authors indicate web site of Roscoff Culture Collection. It was a little bit difficult to find RCC1710 because the distribution was finished (The default setting was "distributed: Yes"). I thought some key information can be summarized in the text or table (e.g. ocean origins). It is also good option to show some details about strains by supplemental information. According to Dr Hagino, RCC 1252 was from Tsugaru straight (Japan) and RCC1710 was captured at off Nagasaki (Japan). Both places are strongly influenced by Tsushima warm current. What is the origin of IAN01? I think it is important facts to keep the reproducibility of the study. Any kind of description would be useful. Can the strain be distributed by request?

P3L12 The water samples were collected after experimental treatment?

P6L28 Why tube show positive relationship with temperature?

P7L15/P9L5 Why the malformed percentage are different among the strains?

P11L35 Control commands can be modified.

P19 Some explanation about empty places might be kindly to readers.

---

## Author Comment (AC1) · 11 Apr 2016

We appreciate the overall positive referee remarks and acknowledge the constructive comments that greatly helped to clarify a number of points and to improve the manuscript.

Below are our detailed responses to the referee's comments, including expected modifications of the manuscript:

COMMENT: The paper would benefit from better structured abstract and additional information in the discussion, namely the authors might want to briefly discuss coccolith function and explain with more detail their formation process, since malformations might occur at different steps of the latter.

[Figure]

REPLY: We added the following: "The coccolith shaping machinery is, besides the ion transport machinery, an essential part of coccolith formation (for an overview see Holtz et al. 2013). The latter commences with heterogeneous nucleation on an organic template, the so called base plate. The nucleation determines crystal axis orientation. Crystal growth proceeds in principle inorganically, with the notable exception that crystal shape is strongly modified by means of a dynamic mould, which essentially consists in the coccolith vesicle shaped by cytoskeleton elements and polysaccharides inside the coccolith vesicle. Malformations can be due to an abnormal base plate which would affect crystal axis orientation, aberrations in the composition or structure of the polysaccharides, and disturbance of cytoskeleton functionality. The latter would most likely also cause a decline in growth rate, which is why this mechanism was disregarded in the case of carbonate chemistry induced malformations (Langer et al. 2011). By the same reasoning, temperature induced malformations might be due to cytoskeleton disturbance, because temperature does also alter growth rate (Fig. 1a). However, it is not straightforward to see why lower than optimum temperature should disturb cytoskeleton functionality (see also Langer et al. 2010). At any rate, coccolith malformations are most likely detrimental to fitness, because malformed coccoliths result in fragile coccospheres, which are regarded as instrumental in coccolithophore fitness (Dixon 1900, Young 1994, Langer and Bode 2011, Langer et al. 2011). One of the many hypotheses concerning function of calcification is that the coccosphere confers mechanical protection (Dixon 1900, Young 1994). After more than a century of research, it still remains the most plausible hypothesis."

COMMENT: Although the temperature range used is physiologically interesting it is broader than expected for the year 2100. Therefore, this should be clear in the discussion.

REPLY: We clarified this. "Although the range of temperatures used here exceeds 2100 projections (IPCC 2013), we not only used it on physiological grounds, but also for ecological reasons. Over the course of the year, coccolithophores

in the North Pacific do experience the whole range of temperatures used here (http://disc.sci.gsfc.nasa.gov/giovanni/, maps in the supplementary material)."

COMMENT: Finally, the manuscript seems to follow reliable experimental procedures and design and is overall well written.

Specific comments and technical corrections: Abstract The abstract would benefit from a statement with the importance and aim of the study as well as a clear conclusion.

REPLY: We added the following. "The global warming debate has sparked an unprecedented interest in temperature effects on coccolithophores. The calcification response to temperature changes reported in the literature, however, is ambiguous. The two main sources of this ambiguity are putatively differences in experimental setup and strain-specificity. In this study we therefore compare three strains isolated in the North Pacific under identical experimental conditions.

In summary, global warming might cause a decline in coccolithophore's PIC contribution to the rain ratio, as well as improved fitness in some genotypes due to less coccolith malformations"

COMMENT: Page 1, line 20 - Replace "E. huxleyi" with " Emiliania huxleyi ".

REPLY: P1, L1. Corrected.

COMMENT: Page 1, line 28 – Increasing PIC production (Figure 1C) was only positively correlated with the percentage of incomplete coccoliths in one strain? Do the authors mean significantly?

REPLY: P1, L6-7. Yes, we have clarified it in the text.

COMMENT: Page 2, line 1-3 – It could be related to time shortage of other steps of the coccolith formation.

REPLY: P1, L7-8. In principle yes, but it is hard to see which steps that could be.

COMMENT: Page 2, line 8 – Clarify the final sentence.

REPLY: P1, L12. We added the following "This clarifies the ambiguous picture featuring in the literature, i.e. discrepancies between PIC:POC-temperature relationships reported in different studies using different strains and different experimental setups"

COMMENT: Introduction Page 3, line 9-10 – Correct and clarify.

REPLY: P1, L18-19. Corrected, now it reads "As a photosynthetic organism, E. huxleyi shifts the seawater carbonate system towards [CO3 2-], but as a calcifier it shifts the seawater carbonate system towards [CO2]."

COMMENT: Page 3, line 10-14 – Long sentence. Improve "... traditionally in particular...".

REPLY: P1, L19-22. Now it reads: Therefore, part of the interest in E. huxleyi derives from its role in the global carbon cycle. Especially extensive blooms (Westbroek et al., 1993; Paasche, 2001), might impact air–sea gas-exchange (Robertson et al., 1994; Buitenhuis et al., 1996).

COMMENT: Page 3, line 14-16 – The sentence should include information concerning stratification, since it is a relevant point in the argument. Moreover, it needs to be clearer.

REPLY: P1, L22-23. Now it reads "Climate change-induced surface water stratification was shown to trigger E. huxleyi blooms (Harada et al., 2012)."

COMMENT: Page 3, line 17-18 – Improve the sentence.

REPLY: P2, L1-2. Now the text reads "The ratio of particulate inorganic carbon (PIC) and particulate organic carbon (POC) influences surface water-atmosphere gas-exchange as well as the composition of matter exported from surface waters to the deep ocean (Ridgwell and Zeebe, 2005; Findlay et al., 2011)."

COMMENT: Page 3, line 19 – Replace "...PIC and the POC production..." with "...PIC

and POC production..." and improve sentence.

REPLY: P2, L3. Done.

COMMENT: Page 4, line 12-13– Clarify the sentence.

REPLY: P2, L15-16. We added "... success of coccolithophores, because different group of organisms might be differently affected by warming and therefore ecological succession patterns, grazing pressure etc., might change".

COMMENT: Material and methods When describing units, like "cells.ml-1", I would remove the ".".

REPLY: This is not present in the BGD published version.

COMMENT: Page 5, line 8 – Explain "alternatively".

REPLY: P2, L28. It refers to the strains synonyms, the text now reads "(RCC1710 −synonym of NG1− and RCC1252 −synonym of AC678 and MT0610E−)".

COMMENT: Page 5, line 10 – Replace "North Sea seawater" with "North Sea water".

REPLY: P2, L30. Done.

COMMENT: Page 5, line 11 – Is it relevant to state "filter cartridges"? Perhaps the filter composition could be more interesting.

REPLY: We added the product information. "Sartobran 300 filter cartridges, Sartorius, Germany)".

COMMENT: Page 5, line 11-12 – Replace "nitrates and phosphates" with "nitrate and phosphate".

REPLY: P2, L31. Done.

COMMENT: Page 6, line 8 and 13– For how long were the samples stored before being measured?

REPLY: We added the information "Samples were stored for less than two months prior to measurement".

COMMENT: Page 6, line 17– Remove "," after TA

REPLY: P3, L18. Done.

COMMENT: Page 6, line 22– I would add a "and" before "calcite".

REPLY: P3, L21. Done.

COMMENT: Page 7, line 2– Storing the samples in a desiccator is important to dry the samples before analysis, correct? How long were they stored in the desiccator?

REPLY: We added the information "Samples were dried for 24 hours in a drying cabinet at 60°C prior to measurement".

COMMENT: Page 8, line 18– Start the sentence with text.

REPLY: P4, L19. Done, now it reads "From 10–30 ml of culture was filtered..."

COMMENT: Results Results could be better organized and more fluid. Reference to effects of the carbonate system could be pooled in one / two paragraphs. Page 10, line 12– For which temperature(s)?

REPLY: P5, L17-18. From 15 to 25°C, this is now specified in the text.

COMMENT: Page 10, line 25– Correct "pgcell-1", "-1" should be superscript.

REPLY: P5, L26. This is corrected in the BGD published version.

COMMENT: Page 11, line 7-9– It is not clear to which carbonate system variable it is being referred here. It would be useful to have a short explanation on the material and methods section concerning the reasoning for following the carbonate system in the experiment. Is it part of your hypothesis? Moreover, the legend of Table 2 should state whether the values are initial, final or averages of the incubations.

REPLY: P6, L1-2. We amended Table 2 and modified as follows. "There was no consistent explanatory variable for cellular PIC, POC, and TPN when analyzing the three strains independently" in Results. And "The seawater carbonate system was monitored because temperature and coccolithophore production alter the system. We employed the dilute batch method (Langer et al. 2013) to minimize production effects" in Material and Methods.

COMMENT: Page 14, line 10-15- The paragraph should be clearer.

REPLY: P7, L18. We agree, now the text reads "Only in strain RCC1710, the percentage of incomplete coccoliths presented a significant increase with temperature (Fig. 6b, Table 4). Higher percentages of incomplete coccoliths in strain RCC1710 were found at 25°C. ANOVA results showed...".

COMMENT: Discussion Page 14, line 19– In spite of being referred in the text, this title does not refer growth rate.

REPLY: P7, L23. We agree, and changed the headline accordingly.

COMMENT: Page 14, line 21– I would replace "versus" with "in relation to".

REPLY: P7, L24. We appreciate the comment, but we have decided to leave it as it is originally written.

COMMENT: Page 14, line 22-23– Specify how many strains and add their isolation sites or areas.

REPLY: P7, L25-26. Here we were just giving examples, it is now clarified.

COMMENT: Page 15, line 6- Why do you only show data for 2 strains and 2 temperatures?

REPLY: P7, L31-32. We wanted to confine the information to the necessary minimum. However, we now added the missing data.

COMMENT: Page 15, line 5- Add reference to support the statement.

REPLY: P7, L30-31. This statement does not require a reference, because this is our own reasoning. We think that this intuition will resonate strongly with many readers, so we regard it as a literary device to improve readability.

COMMENT: Page 15, line 6- Percentage of incomplete coccoliths is higher under 20-25°C than 10-15°C in RCC1252. Thus, if one would consider the temperatures used to determine the coccolith production time, both strains would show similar trends, or not? The lack of significance precludes a strong conclusion concerning this parameter.

REPLY: We do not understand the reasoning of this comment.

COMMENT: Page 15, line 10- What about coccolith mass?

REPLY: We discussed this in page 8, lines 4-6.

COMMENT: Page 16, line 14- The authors refer a previous study (Sett et al., 2014) that showed a different relation between PIC:POC and temperature in the introduction section. This should be referred in the discussion section.

REPLY: Page 8, line 21. We now cite Sett et al. 2014.

COMMENT: Page 17, line 6- Add references supporting "... most strains live at sub-optimal temperatures in the field.".

REPLY: Page 8, line 33. We added the references.

COMMENT: Page 17, line 10-13- Clarify what is meant with "short timescales".

REPLY: Page 9, line 3. Done. "...might even affect surface water carbonate chemistry on short timescales, i.e. one year".

COMMENT: Page 17, line 19-20- Specify which artificial conditions could play a role in producing malformations of coccoliths.

REPLY: We added the following. "... artificial conditions such as cell densities of 10ˆ6

cells/ml, cells sitting on the bottom of the culture flask, stagnant water, confinement in a culture flask. . ..."

COMMENT: Page 18, line 13-15- One of the strains tested in this study is considered to belong to the warm-water group while the others not. Why is it referred in the text and how can it affect the observed responses to increasing temperatures?

REPLY: As explained in lines 23-25 page 9, RCC1252 should show the same response pattern as RCC1238 and BT-6, if the warm water group strains share common features. Since it does not show the same response, the fact that RCC1252 belongs to the warm water group does not mean anything in this context.

COMMENT: References Young and Westbroek, 1991 cited on page 15 line 21 is missing in the references. Several references have typos, namely missing italics, incorrect formatting of the 2 of CO2 (should be subscript) and accents.

REPLY: The missing reference is already added in the BGD published version. We will correct all the other typos.

COMMENT: Page 30, line 7– Incorrect date, it should be 1993.

REPLY: This error is not present in the BGD published version.

COMMENT: Legends Table 5 and Figure 2, 3, 4- Species should be in italic and presented in a consistent form, either E. huxleyi or Emiliania huxleyi.

REPLY: We agree, the species names in italics are already correct in the BGD published version.

COMMENT: Figure 1, page 37, line 4- missing a space between "(E) and".

REPLY: Typo already corrected in the BGD published version.

COMMENT: Tables Table 5- The percentage of incomplete coccoliths is considered a strain-specific response, why? The author's should clarify the choice of responses in

the legend of the table. Was it based on significance?

REPLY: Yes it was based on significance. We have now clarified this point in the caption of the table.

COMMENT: Figures Figure 1, 5 and 6 are hard to analyze due to size / resolution. Labels (A, B. . .) of the figures and text should follow the same formatting. Figures that do not start with 0, should show an interruption on the axis. Finally, when the unit is in the axis title it is not necessary to have it close to the numbers (see for instance Figure 6).

REPLY: The resolution of figures and the labels are already corrected in the BGD published version. According to your comment, we have removed the unnecessary "%" symbols.

---

## Author Comment (AC2) · 11 Apr 2016

We appreciate the overall positive referee remarks and acknowledge the constructive comments that greatly helped to clarify a number of points and to improve the manuscript.

Below are our detailed responses to the referee's comments, including expected modifications of the manuscript:

COMMENT: I would like to recommend authors to show the reason why the three strains are chosen. This species widely distributed from tropical to sub-polar region. If the strains originated from cooler environment were chosen, the opposite relationships may be shown between production and temperature. The environmental adaptation might be important to include consideration of the result. The other ecological information of them are also valuable if they are available.

REPLY: We agree and have now included the information on strain choice. The new text reads "Clonal cultures of Emiliania huxleyi were obtained from the Roscoff Culture Collection. We selected 3 strains of E. huxleyi, two from the Japanese coast in the North Pacific Ocean (RCC1710 –synonym of NG1– and RCC1252 –synonym of AC678 and MT0610E–) and a third strain from the same region but of unknown exact origin and strain name, named here IAN01. Strain RCC1710 was collected off Nagasaki at Tsushima straight (Japan) and RCC1252 at Tsugaru straight (Japan), both places are strongly influenced by the Tsushima warm current. Additional information about the strain RCC1252 can be found at: http://roscoff-culture-collection.org/." We selected the strains because they were collected in the tropical North Pacific and we wanted to test different strains from a single area in order to assess the plasticity within strains originating from a particular environmental setting. We also agree that strains from a cooler environment are likely to respond differently to the temperature range tested here, e.g. the optimum will probably be a lower temperature. We now include this point. In the introduction "We selected three strains of E. huxleyi from a single area, the Japanese coast in the North Pacific Ocean, in order to assess the plasticity within strains originating from a particular environmental setting" and in the Discussion "All three E. huxleyi strains investigated here displayed similar growth rate versus temperature relationships, with an optimum at 20-25°C (Fig. 1a). This is a typical range for many E. huxleyi strains (Watabe and Wilbur, 1966; Van Rijssel and Gieskes, 2002; Sorrosa et al., 2005; De Bodt et al., 2010; Langer et al., 2009). We expect that strains isolated e.g. in the Arctic will have a lower temperature optimum, though." We noticed too late that the strain IAN01 was wrongly labelled at an early stage of the study. Tracing back this label we could do no more than infer that it was isolated in the same area as the other strains, but we could not unambiguously identify its name. That is why we gave it a new name. For our study it is important that IAN01 comes from the same area as the other strains. So it is unfortunate, but not critical, that we do not know its real name.

[Figure]

COMMENT: P2L30 The authors indicate web site of Roscoff Culture Collection. It was a little bit difficult to find RCC1710 because the distribution was finished (The default setting was "distributed: Yes"). I thought some key information can be summarized in the text or table (e.g. ocean origins). It is also good option to show some details about strains by supplemental information. According to Dr Hagino, RCC 1252 was from Tsugaru straight (Japan) and RCC1710 was captured at off Nagasaki (Japan). Both places are strongly influenced by Tsushima warm current. What is the origin of IAN01? I think it is important facts to keep the reproducibility of the study. Any kind of description would be useful. Can the strain be distributed by request?

REPLY: We now included the information on the strains (see reply to previous comment).

COMMENT: P3L12 The water samples were collected after experimental treatment?

REPLY: Yes, during the harvesting. We have clarified this in the text "During the harvesting, samples for total alkalinity (TA) measurements were sterile-filtered (0.2 $\mu$m pore size) and stored in 25 ml borosilicate flasks at 4°C until measurements."

COMMENT: P6L28 Why tube show positive relationship with temperature?

REPLY: We clarified that point "The positive relationship of the mean tube width with temperature reflects the increased coccolith calcite quota at higher temperature. Coccolith mass and coccolith size are positively correlated. Why coccolith mass or size should increase with temperature cannot be decisively answered based on our data."

COMMENT: P7L15/P9L5 Why the malformed percentage are different among the strains?

REPLY: We clarified that point "The fact that the base level of malformations in cultured coccolithophores differs between species and strains (and also varies with time) has been recognized for many years and is now well documented (e.g., Langer and Benner, 2009; Langer et al., 2011, 2013). Also the response of the morphogenetic machinery

to environmental factors is strain-specific (Langer et al., 2011; Oviedo et al., 2014). We have currently not enough accessory information to formulate a hypothesis why exactly one strain differs from another. That fact that they do indeed differ, however, probably reflects the high genetic diversity in E. huxleyi."

- Langer, G. and Benner, I.: Effect of elevated nitrate concentration on calcification in Emiliania huxleyi, Journal of Nannoplankton Research, 30, 77–80, http://epic.awi.de/22502/, 2009.

- Langer, G., Probert, I., Nehrke, G., and Ziveri, P.: The morphological response of Emiliania huxleyi to seawater carbonate chemistry changes: an inter-strain comparison, Journal of Nannoplankton Research, 32, 29–34, 2011.

- Langer, G., Oetjen, K., and Brenneis, T.: On culture artefacts in coccolith morphology, Helgoland Marine Research, 67, 359–369, doi:10.1007/s10152-012-0328-x, http://link.springer.com/10.1007/s10152-012-0328-x, 2013.

- Oviedo, A. M., Langer, G. & Ziveri, P.: Effect of phosphorus limitation on coccolith morphology and element ratios in Mediterranean strains of the coccolithophore Emiliania huxleyi, Journal of Experimental Marine Biology and Ecology. 459, 105-113, 2014.

COMMENT: P11L35 Control commands can be modified.

REPLY: Thanks for noticing the typo, it has been corrected.

COMMENT: P19 Some explanation about empty places might be kindly to readers.

REPLY: We have now added an explanation. "Growth rate and cellular PIC, POC, and TPN content and production of the three strains of E. huxleyi at different temperatures. Standard deviation of the triplicates in parentheses. Measured growth rates for extra temperatures from the pre-experiments are included, but PIC, POC and TPN were not measured for these temperatures."